# Feasibility of Schema Therapy for Recurrent Depression in a Disaster Relief Worker with Prior Post-Traumatic Stress Disorder Treatment Using Prolonged Exposure Therapy

**DOI:** 10.3390/bs14121156

**Published:** 2024-12-02

**Authors:** Arinobu Hori, Michio Murakami, Fumiyo Oshima, Remco van der Wijngaart

**Affiliations:** 1Department of Psychiatry, Hori Mental Clinic, 106 Gozen-no-uchi, Kashima, Kashima-ku, Minamisoma 979-2335, Fukushima, Japan; 2Department of Neuropsychiatry, Fukushima Medical University School of Medicine, 1Hikariga-oka, Fukushima 960-1295, Fukushima, Japan; 3Center for Infectious Disease Education and Research (CiDER), Osaka University, Techno Alliance C209, 2-8 Yamadaoka, Suita 565-0871, Osaka, Japan; michio@cider.osaka-u.ac.jp; 4Research Center for Child Mental Development, Chiba University, 1-8-1 Inohana, Chuouku 260-8670, Chiba, Japan; c21ujsw35117c@faculty.gs.chiba-u.jp; 5Dutch Institute for Schema Therapy, Van Genderen Opleidingen BV, Burgemeester Ceulenstraat 102, 6212 CV Maastricht, The Netherlands; remcovanderwijngaart@gmail.com

**Keywords:** post-traumatic stress disorder (PTSD), disaster relief worker, prolonged exposure (PE) therapy, the Great East Japan Earthquake (GEJE), schema therapy

## Abstract

This report presents the follow-up treatment course of a previously published case that demonstrated the effectiveness of prolonged exposure (PE) therapy for a disaster relief worker. The patient, a municipal employee in Fukushima Prefecture, developed post-traumatic stress disorder (PTSD) and mood disorders after the 2011 Great East Japan Earthquake and subsequent disasters. This follow-up focuses on the period from 2021 to early 2024, during which the patient experienced symptom recurrence after his father’s death. This event revealed psychological patterns similar to his disaster-related responses. Schema therapy was introduced to address over-adaptive work behaviors and vulnerabilities in relationships, identified as relapse risk factors. Combined with antidepressants, schema therapy achieved sustained improvement. This longitudinal perspective demonstrates schema therapy’s effectiveness in addressing underlying vulnerabilities when symptoms re-emerge after trauma-focused treatment. The findings underscore how initial trauma responses may interact with subsequent life events and suggest schema therapy’s potential as a secondary intervention for disaster relief workers facing complex challenges.

## 1. Introduction

Mental health care for disaster relief workers has become increasingly important as natural and technological disasters continue to affect communities worldwide. These workers face unique challenges, being both disaster survivors and professional responders [1,2]. Following the Great East Japan Earthquake (GEJE) of 2011, surveys revealed significant mental health impacts among public employees in affected areas. In Fukushima Prefecture, where workers faced compound disasters including earthquake, tsunami, and nuclear accident, 17.9% of public employees exhibited depression [3]. Similarly, in Miyagi Prefecture, 6.7% of local municipality and medical workers required intervention for PTSD-related symptoms [4].

While disaster-related mental health research often focuses on immediate trauma responses, long-term follow-up studies reveal complex interactions between disaster-related PTSD and subsequent life events. Disaster relief workers, in particular, may experience delayed or recurring symptoms triggered by personal life events that resonate with their disaster-related experiences. These cases require careful attention to both the initial trauma processing and underlying vulnerabilities that may influence long-term recovery. Understanding how disaster-related trauma responses interact with later life challenges is crucial for developing comprehensive treatment approaches for this population. This perspective becomes particularly relevant when considering cases where initial treatment gains are not maintained despite successful trauma-focused interventions.

Our previous case report [5] documented a municipal employee who developed post-traumatic stress disorder (PTSD) and depression 3.5 years after the GEJE. The worker had experienced multiple traumatic events, including retrieving bodies of acquaintances and managing evacuations amid residents’ distress, while personally affected by nuclear evacuation orders. The case demonstrated a complex clinical course: initial improvement with rest and medication was followed by recurrence, necessitating prolonged exposure (PE) therapy [6,7]. The PE intervention proved effective, enabling the patient to return to work and maintain stability even during subsequent disasters, including a major flood and the coronavirus disease (COVID-19) pandemic. However, post-treatment conference discussions with PE therapy specialists raised concerns about potential insufficient emotional processing and over-adaptation to professional demands.

While evidence-based trauma-focused therapies, particularly PE, have shown effectiveness in treating PTSD among disaster relief workers, the phenomenon of symptom recurrence despite initial improvement warrants attention. This pattern suggests the influence of underlying vulnerability factors, including work-related over-adaptation common among disaster relief workers [8,9]. Such over-adaptation, while initially functional for maintaining high-quality support activities in overwhelmingly demanding conditions, may mask personal challenges and emotional processing needs. Disaster relief workers often must suppress their own negative emotions, such as grief and distress, to continue providing essential services in affected communities.

Schema therapy, developed as an integrative approach for chronic psychological issues, offers a promising framework for addressing these underlying vulnerabilities [10,11]. Through its focus on modifying early maladaptive schemas (EMS)—pervasive patterns comprising memories, emotions, cognitions, and bodily sensations about oneself and relationships—this approach has demonstrated effectiveness not only for personality disorders [12,13,14,15] but also for various mental health conditions [16]. While various evidence-based approaches effectively address trauma symptoms, schema therapy uniquely focuses on how early experiences shape both trauma responses and coping patterns. This becomes especially pertinent in cases where initial trauma-focused treatments achieve symptom reduction but underlying vulnerability patterns persist. The approach’s emphasis on understanding and modifying long-standing coping modes aligns particularly well with the needs of disaster relief workers, who often develop specific professional coping patterns that may interact with personal vulnerability factors [11]. However, its application in cases of disaster relief workers who experience symptom recurrence after successful trauma-focused treatment remains unexplored.

This paper presents a follow-up to our previous case report [5], describing the subsequent course of the same municipal employee who, despite initial improvement with PE therapy, experienced symptom recurrence following a personal loss. Through examining this case, we explore the potential value of schema therapy as a secondary intervention when initial trauma-focused treatment gains are not maintained, particularly in addressing the complex interplay between professional responsibilities and personal vulnerabilities in disaster relief workers facing prolonged challenges in disaster-affected communities.

## 2. Case Presentation

This case report follows a naturalistic, longitudinal observation approach, documenting the clinical course and therapeutic interventions as they occurred in a clinical setting. Clinical data was collected through detailed session notes, patient’s self-reported experiences and symptoms, treatment response documentation, and regular follow-up assessments during outpatient visits, supplemented by family member observations when available. The clinical course spans approximately 13 years, with particular focus on the period from 2021 to 2024. Treatment decisions throughout this period were made based on clinical necessity rather than research protocols.

### 2.1. Initial Presentation and Background

The patient was in his 40s when he first visited our institution. His final diagnosis was PTSD and recurrent depression. During the GEJE and nuclear accident, he lived with his wife and daughters. He experienced the earthquake and tsunami firsthand but continued supporting residents under severe conditions as a city hall employee, often facing harsh criticism from the citizens. Although his family temporarily evacuated to other regions, he did not. Table 1 presents the complete clinical course of this case.

### 2.2. Course of Illness and Initial Treatment

Approximately 3.5 years after the earthquake, PTSD and depression became apparent after he watched the news of another disaster in Japan, necessitating a leave of absence. Despite psychiatric treatment and symptom improvement with medication and rest, relapses occurred upon returning to work. During this time, episodes of elevated mood, spending sprees, and alcohol consumption were observed. He was referred to our medical institution by his previous doctor after a 4-year treatment period, presenting on 20 mg of escitalopram and 30 mg of mirtazapine daily.

### 2.3. PE Therapy Phase

Approximately 7.5 years after the earthquake, 10 sessions of PE therapy [6] were conducted. These sessions primarily involved detailed recounting of the days immediately following the disaster, focusing on the fear experienced during the tsunami and the retrieval of several bodies, including those of acquaintances. Emotional processing also included discussing thoughts about his deceased acquaintance. The treatment resulted in symptom improvement, and the patient returned to work, maintaining stability with monthly outpatient visits.

### 2.4. Subsequent Challenges and Treatment

At 8.5 years post-earthquake, when his community experienced large-scale flooding, the patient managed evacuation shelters effectively despite minimal sleep. However, symptoms relapsed afterward, requiring a 1-month leave. Two additional exposure sessions addressing unresolved memories of resident criticism led to recovery and work resumption.

### 2.5. Relapse and Schema Therapy

Approximately 10.5 years after the compound disaster, following his father’s health deterioration and subsequent death, the patient’s depression relapsed. As PE therapy was presumed ineffective for this non-trauma-related trigger, schema therapy [10,11] was proposed 11 years and 10 months after the initial disaster. The therapy consisted of approximately 60 sessions over 1 year and 4 months, with sessions initially weekly and later bi-weekly. The patient successfully returned to work after the 46th session.

### 2.6. Current Status

At 12 years and 4 months post-disaster, the patient maintains regular employment while continuing outpatient visits. His current medication regimen includes escitalopram 10 mg, lurasidone 40 mg, and quetiapine 25 mg daily.

## 3. Schema Therapy Process and Outcomes

### 3.1. Developmental History and Early Experiences

The schema therapy intervention, conducted over 60 sessions, encompassed five main components. First, a thorough examination of the patient’s developmental history revealed significant childhood experiences, particularly regarding his relationship with his father. The father was described as work-focused and emotionally distant, maintaining a calm but intimidating presence. While rarely engaging with the patient, he would become severely angry over academic issues. Due to limited parental interaction, the patient spent considerable time with his grandparents. A notable pattern of compliant responses to school bullying was also identified.

### 3.2. Core Emotional Needs and Schema Identification

The second component involved psychological education on core emotional needs and EMSs [10,11]. The therapy framework addressed five domains of core emotional needs:①secure attachment to others②autonomy, competence, and identity sense③freedom to express valid needs and emotions④spontaneity and play⑤realistic limits and self-control.

Through this framework, several schemas were identified across domains, including abandonment, emotional deprivation, and social isolation in Domain 1, and vulnerability to harm, enmeshment, and failure in Domain 2. Domain 3 revealed subjugation, self-sacrifice, and approval-seeking schemas, while Domain 4 showed perfectionism and punitiveness schemas. Table 2 presents three key early maladaptive schemas identified in this patient.

**Table 2 behavsci-14-01156-t002:** Explanation for 3 early developmental schemas (created by the authors referring to the text of [10]).

Abandonment	The perceived instability or unreliability of those available for support and connection. It involves the sense that significant others cannot continue providing emotional support, connection, strength, or practical protection.
Enmeshment/undeveloped self	Excessive emotional involvement and closeness with multiple significant others at the expense of full individuation or normal social development. Often involves the belief that at least one of the enmeshed individuals cannot survive or be happy without the constant support of the other.
Subjugation	Excessive surrendering of control to others out of fear of anger, retaliation, or abandonment. Usually, it involves the perception that one’s desires, opinions, and feelings are invalid or unimportant to others.

### 3.3. Mode Work and Experiential Techniques

The third component introduced the concept of “modes” to understand recurring patterns in the patient’s life. Modes, defined as “states that manifest from the interaction of activated EMS and coping styles at a given moment” [11], include child, critic, coping, and healthy adult modes. Through experiential techniques, the therapy addressed situations activating the vulnerable child mode, particularly using imagery exercises where a competent caregiver offers emotional support. When punitive and demanding critic modes emerged, interventions focused on strengthening healthy adult modes to make reasonable protests to inner critics. This process helped the patient understand how early experiences of submission to his father’s harsh demands influenced his adult tendency to comply with demanding situations.

### 3.4. Identification and Modification of Maladaptive Coping

The fourth component focused on identifying and modifying the patient’s distinctive maladaptive coping mode, termed the “armor mode.” This mode manifested as emotional disconnection, treating events as tasks, and focusing solely on task management. Through therapy, the patient recognized how this mode led his family to feel unheard and ignored. Working on this awareness led to improved emotional commitment in therapy and enhanced communication with his spouse, including processing shared memories of the GEJE. Notably, the patient’s workplace behavior also transformed. Previously viewed as someone who “took on work and responsibilities alone and then suddenly took leave, claiming illness”, he began seeking supervisor advice more proactively when facing difficulties, a change supported by research showing the benefits of effective supervisor communication in disaster-affected workplaces [17].

### 3.5. Therapeutic Process and Clinical Progress

The schema therapy intervention demonstrated distinct progression through three phases, each characterized by specific therapeutic focuses and observable changes in the patient’s engagement and responses.

In the initial phase (Sessions 1–21), therapy concentrated on psychoeducation and cognitive understanding of childhood trauma. This period focused on gathering developmental history and introducing schema therapy concepts and cognitive-behavioral techniques. A significant finding was the identification of repeated coercive paternal interventions, often involving physical punishment, particularly regarding academic achievement. The patient showed a marked reluctance to confront these memories, requiring careful therapeutic support. This phase concluded with the patient achieving cognitive understanding of the connection between these experiences and his EMSs, particularly abandonment, isolation, subjugation, and punitive schemas.

The intermediate phase (Sessions 22–40) focused on addressing childhood trauma through experiential techniques. During this period, the patient repeatedly processed memories of paternal violence and intimidation, experiencing significant fear responses. Therapeutic interventions primarily utilized imagery exercises, particularly those involving a nurturing parent figure supporting the vulnerable child mode. By the end of this phase, the patient demonstrated an increased capacity to challenge the internalized punitive parent image from a healthy adult perspective in imagery work.

The final phase (Sessions 41–60) centered on identifying and modifying a distinctive coping mode, termed the “armor mode”, characterized by emotional disconnection and task-focused behavior. This mode was recognized as a childhood adaptation for managing academic demands under paternal intimidation. Therapeutic work focused on bypassing this mode to access authentic emotions while maintaining adaptive functioning. The later sessions integrated these insights through repeated imagery work and cognitive reformulation, including processing grief over paternal loss and addressing maternal relationships. This phase demonstrated the patient’s enhanced emotional awareness while functioning effectively in professional settings.

## 4. Discussion

The present case highlights several important aspects of treating disaster relief workers with complex trauma histories. First, this case demonstrates the compound and prolonged nature of disaster-related mental health challenges. As noted by Maeda et al. [8] and Hori et al. [9], the 2011 GEJE had a severe, long-term impact on survivors’ mental and physical health. Shigemura et al.’s systematic review [18] found that 8.3–62.6% of survivors experienced nonspecific psychological distress, 12–52% had depressive symptoms, and 10.5–62.6% showed PTSD symptoms. The situation was further complicated by subsequent disasters, including floods, earthquakes, and the COVID-19 pandemic, creating a continuous challenge for mental health interventions. This complexity is reflected in the high treatment discontinuation rate (32.9%) reported by a regional psychiatric facility [19].

Second, this case illustrates the unique position of disaster relief workers as both survivors and professional responders. Previous research by Garbern [1] and Naushad [2] identified disaster relief workers as a high-risk group for mental health disorders, including depression and PTSD. Our patient’s experience aligns with Rahnama’s research [20] on bipolar coping strategies among healthcare workers during the COVID-19 pandemic. His disaster relief activities could be interpreted as positive protective coping aimed at protecting colleagues and the local community, rather than purely symptomatic behavior.

Third, our case demonstrates both the effectiveness and limitations of PE therapy in treating disaster-related PTSD. PE therapy successfully addressed the patient’s PTSD symptoms and enhanced resilience when facing subsequent natural disasters, including floods and earthquakes. However, the patient remained vulnerable to family-related psychological stressors, as evidenced by his reaction to his father’s illness and death. This limitation highlights the need for comprehensive treatment approaches that address both acute trauma symptoms and underlying psychological vulnerabilities.

Fourth, and perhaps most significantly, this case provides insights into the role of schema therapy in addressing persistent vulnerabilities after successful trauma-focused treatment. The patient’s “armor mode”, characterized by emotional disconnection and task-focused coping, represents a maladaptive pattern common among individuals with avoidant-dependent traits. While this coping style initially enabled continued functioning in demanding situations, it ultimately hindered emotional processing and relationship development. As noted in the schema therapy literature [11], such coping modes can interfere with therapeutic progress. A key advantage was schema therapy’s capacity to address childhood trauma—in this case, paternal abuse—which had led to the development of the “armor mode”. This coping mechanism impeded recognition of emotional needs and potentially reduced the impact of previous therapeutic interventions. Patient satisfaction tends to be high as individuals experience increased fulfillment of core emotional needs. However, significant challenges exist, including the extended duration of treatment and the limited availability of qualified therapists and facilities. From a cost-effectiveness perspective, schema therapy may be most appropriate for cases involving early developmental issues that complicate trauma responses, rather than as a first-line treatment for PTSD without such underlying patterns. The successful implementation, in this case, suggests its potential value as a secondary intervention when initial trauma-focused treatment gains are not maintained, enabling the patient to develop more adaptive coping strategies and improve both workplace and family relationships (Table 3).

## 5. Limitations

Several limitations of this case report should be acknowledged. First, as this is a single case study from a small clinical practice, the generalizability of our findings is inherently limited. Second, due to the naturalistic observation approach, we did not conduct systematic psychometric assessments, which could have provided quantitative evidence of symptom changes and treatment effectiveness. Third, while schema therapy showed effectiveness in this case, the extended duration of treatment (60 sessions over approximately 16 months) and the need for specialized training may limit its practical implementation in resource-limited settings. Fourth, our observations of improved workplace functioning and relationships primarily relied on patient self-reports and family observations, without standardized measures of functional improvement. Finally, the specific context of this case—a disaster relief worker in Fukushima following the GEJE—may limit the applicability of our findings to other populations or settings.

## 6. Conclusions

These findings suggest that comprehensive treatment planning for disaster relief workers should consider both immediate trauma-focused interventions and longer-term approaches addressing underlying vulnerability factors. Future research might explore the optimal timing and sequencing of these interventions, as well as specific adaptations for disaster relief workers who face ongoing community challenges while managing their own recovery process.

## Figures and Tables

**Table 1 behavsci-14-01156-t001:** Clinical Course of a Disaster Relief Worker Following the Great East Japan Earthquake (GEJE): 2011–2024.

Phase	Year	Events and Interventions
Initial Phase (2011–2014)	2011	GEJE, tsunami, and nuclear accident occurred Continued working as disaster relief worker Maintained function despite stress
	2014	PTSD and depression onset triggered by news of another disaster Intervention: Rest and medication Outcome: Temporary improvement
Treatment Phase I (2014–2020)	2014-2018	Recurrent depression Intervention: Psychiatric treatment Outcome: Partial response with relapses
	2018	Referral to our institution Persistent PTSD symptoms Intervention: 10 sessions of PE therapy Outcome: Significant improvement
	2019	Local flooding disaster Temporary symptom exacerbation Intervention: 2 additional PE sessions Outcome: Recovery and return to work
Treatment Phase II (2021–2024)	2021	Father’s illness and death Depression relapse Intervention: Medical leave Outcome: Limited improvement
	2023	Persistent symptoms Intervention: Schema therapy initiated (60 sessions) Outcome: Progressive improvement
	2024	Work resumption (after session 46) Improving symptoms Outcome: Maintained improvement

Abbreviations: PTSD: post-traumatic stress disorder; GEJE: Great East Japan Earthquake; PE: Prolonged Exposure.

**Table 3 behavsci-14-01156-t003:** Comparison between prolonged exposure (PE) and schema therapy.

Aspect	Prolonged Exposure Therapy	Schema Therapy
Focus	Directly targets PTSD symptoms related to specific traumatic events	Addresses underlying maladaptive schemas and coping styles that contribute to chronic psychological distress, including trauma-related issues
Approach	Behavioral: Involves repeated exposure to trauma-related memories, thoughts, and situations to facilitate emotional processing	Integrative: Combines elements of cognitive, behavioral, and experiential therapies to modify maladaptive schemas
Duration	Typically, 6–20 weekly sessions, each lasting 90 min	Usually longer term, ranging from 1 to 3 years, with weekly sessions
Efficacy	Highly effective for PTSD, with strong empirical support	Effective for complex trauma-related issues, personality disorders, and chronic psychological difficulties; growing evidence-base
Limitations	May not address underlying schema-level beliefs or coping styles that contribute to chronic psychological distress	Requires longer time commitment and may not focus as specifically on PTSD symptoms as PE

PTSD: post-traumatic stress disorder.

## Data Availability

The original contributions presented in this study are included in the article. Further inquiries can be directed to the corresponding author.

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
