# Peer review of "Feasibility of Schema Therapy for Recurrent Depression in a Disaster Relief Worker with Prior Post-Traumatic Stress Disorder Treatment Using Prolonged Exposure Therapy"

_behavsci, 2024, doi:10.3390/bs14121156_

Round 1

Reviewer 1 Report

Comments and Suggestions for Authors

This article appears to report on a follow up on a previous case study regarding PTSD and PE. It highlights clinical care of an important group of patients – disaster relief victims and addresses a gap in the literature of treating these patients. Overall, it is a very well written article but require some minor edits. Please see below.

General

This article is missing and needs to highlight why this single case study was so important to be followed over this period of time. It does a well job explain “how” it was followed but lacks the “why” this particular case was followed. This is highly crucial in abstract as well.

Tables that explain a timeline can be considered re-formatted into a timeline to be clear digestible for readers.

Introduction/Abstract

In the introduction and abstract section it needs be clarified what follow up time this new paper is capturing in data in regards to the previous published paper to be easily accessible to readers.

Data Reporting

Are they any psychometric data that tracks patient’s symptoms (anxiety measures, depression measures, PTSD measures)? This gives reviewers finite data.

Did any of his providers meet as a treatment team to discuss his care?

Line 84: The articles mentions that the patient was in his 40s – are there any specific data in regards to age as he progresses through the treatments?

Author Response

This article appears to report on a follow up on a previous case study regarding PTSD and PE. It highlights clinical care of an important group of patients – disaster relief victims and addresses a gap in the literature of treating these patients. Overall, it is a very well written article but require some minor edits. Please see below.

Response: Thank you for your valuable feedback on our manuscript. We appreciate your recognition of our work's contribution regarding PTSD treatment for disaster relief victims. We carefully address all the suggested revisions to enhance the manuscript's clarity and impact, particularly focusing on better articulating the rationale for this long-term follow-up case study.

General

This article is missing and needs to highlight why this single case study was so important to be followed over this period of time. It does a well job explain “how” it was followed but lacks the “why” this particular case was followed. This is highly crucial in abstract as well.

Response: Thank you for highlighting the need to better explain the importance of following this specific case study over an extended period. In response, we have revised both the Introduction and the Abstract to address your concerns.

In the Introduction, we have added a new second paragraph to emphasize the unique insights this case provides into the interplay between disaster-related PTSD and subsequent life events. This paragraph highlights how this case exemplifies the delayed or recurring symptoms triggered by personal life events that resonate with disaster-related experiences, making it particularly significant for understanding the long-term challenges faced by disaster relief workers. It also underscores the relevance of this perspective in situations where initial trauma-focused treatment gains are not maintained.

We have also rewritten the Abstract to reflect this rationale clearly. The revised Abstract now specifies why this case warrants long-term follow-up, emphasizing the unique contribution of the case to understanding the interaction between initial trauma responses and later life challenges. It also demonstrates the added value of schema therapy as a secondary intervention when initial trauma-focused treatments no longer sustain recovery.

These changes aim to address the critical “why” question you raised, ensuring the significance of this study is clearly articulated for the reader.

Tables that explain a timeline can be considered re-formatted into a timeline to be clear digestible for readers.

Response: Thank you for this helpful suggestion regarding the timeline presentation. We agree that a visual timeline would enhance reader comprehension and will revise the current table format into a clear, chronological timeline visualization as below:

Table 1. Clinical Course of a Disaster Relief Worker Following the Great East Japan Earthquake (GEJE): 2011-2024
Phase Year  Events and Interventions
Initial Phase 2011 ・GEJE, tsunami, and nuclear accident occurred
(2011-2014)    ・Continued working as disaster relief worker
    ・Maintained function despite stress 
  2014 ・PTSD and depression onset triggerd by news of another disaster
    ・Intervention: Rest and medication
    ・Outcome: Temporary improvement
Treatment Phase I 2014-2018 ・Recurrent depression 
(2014-2020)   ・Intervention: Psychiatric treatment
    ・Outcome: Partial response with relapses
  2018 ・Referral to our institution
    ・Persistent PTSD symptoms
    ・Intervention: 10 sessions of PE therapy
    ・Outcome: Significant improvement
    Local flooding disaster
  2019 ・Temporary symptom exacerbation
    ・Intervention: 2 additional PE sessions
    ・Outcome: Recovery and return to work
Treatment Phase II 2021 ・Father's illness and death
(2021-2024)   ・Depression relapse
    ・Intervention: Medical leave
    ・Outcome: Limited improvement
  2023 ・Persistent symptoms
    ・Intervention: Schema therapy initiated (60 sessions)
    ・Outcome: Progressive improvement
  2024 ・Work resumption (after session 46)
    ・Improving symptoms
    ・Outcome: Maintained improvement 
Abbreviations: 
PTSD: post-traumatic stress disorder 
GEJE: Great East Japan Earthquake 
PE: Prolonged Exposure

Introduction/Abstract

In the introduction and abstract section it needs be clarified what follow up time this new paper is capturing in data in regards to the previous published paper to be easily accessible to readers.

Response: Thank you for your valuable feedback regarding the need to clarify the follow-up time frame captured in this study. In response, we have made the following adjustments:

Abstract:

We have revised the Abstract to explicitly state the follow-up period captured in this study. The Abstract now mentions that the follow-up focuses on the period from 2021 to early 2024, providing readers with a clear temporal framework. This adjustment ensures that the scope of this report is immediately accessible and aligns with the detailed timeline in the Case Presentation section.

Case Presentation Section:

The specific timeline, including the follow-up period from 2021 to 2024, is provided in detail within the Case Presentation section. This approach allows for a comprehensive account of the patient’s treatment course and ensures that readers seeking detailed chronological data can easily locate it. We recognize that this choice might differ from expectations for the Introduction or Abstract, and we sincerely apologize for any confusion this may have caused.

We hope these clarifications address your concerns by balancing the accessibility of the Abstract with the detailed timeline provided in the Case Presentation section. We deeply appreciate your constructive comments, which have helped us enhance the clarity and structure of our manuscript.

Data Reporting

Are they any psychometric data that tracks patient’s symptoms (anxiety measures, depression measures, PTSD measures)? This gives reviewers finite data.

Response: Thank you for your valuable comment regarding the inclusion of psychometric data to track the patient's symptoms. We regret to inform you that such data is not available for this case report. As this case originated from a small clinical practice setting, the primary focus was on delivering individualized patient care rather than systematically collecting data for research purposes. This limitation reflects the realities of clinical practice in resource-limited environments, where psychometric assessments may not be routinely implemented.

While psychometric data were not collected due to the limitations of a small clinical practice setting, this case report offers qualitative insights into the implementation and outcomes of evidence-based therapies in real-world conditions. These findings contribute to understanding the practical challenges and successes of therapy in resource-limited settings, where individualized care often takes precedence.

We have explicitly addressed this limitation in our newly created "Limitations" section (shown in red), stating: "Second, due to the naturalistic observation approach, we did not conduct systematic psychometric assessments, which could have provided quantitative evidence of symptom changes and treatment effectiveness."

We sincerely appreciate your understanding of these limitations and your consideration of the unique context in which this report was developed. Your feedback highlights the importance of integrating more systematic psychometric evaluations in future work, and we will strive to incorporate this approach where feasible in future clinical reports.

Did any of his providers meet as a treatment team to discuss his care?

Response: Thank you for raising this important question regarding the use of treatment team meetings in this case. We acknowledge the critical role that multidisciplinary team discussions often play in comprehensive patient care. However, in this particular case, the primary therapeutic interventions—Prolonged Exposure (PE) therapy and schema therapy—were conducted as individual psychotherapies, which do not typically require formal team meetings as part of their implementation.

While collaborative discussions can enhance patient care in many clinical settings, this report focuses on the psychotherapeutic processes and outcomes within the context of one-on-one treatment. The absence of team meetings does not diminish the rigor of the therapies provided, as these approaches were delivered according to established evidence-based practices.

We appreciate your thoughtful feedback and recognize the broader importance of team-based approaches in mental health care. In this case, however, the individual-focused nature of the therapies reflects the specific needs and structure of the clinical context in which the patient was treated.

Thank you again for your important comment, which has allowed us to clarify the scope and focus of this case report.

Line 84: The articles mentions that the patient was in his 40s – are there any specific data in regards to age as he progresses through the treatments?

Response: Thank you for your insightful comment regarding the patient's age and its potential relevance to his treatment process. While protecting patient privacy limits our ability to provide specific age details, Table 1 presents the chronological progression of events and treatment phases over the 12-year period. As noted, the patient was a municipal employee in his 40s with significant professional and family responsibilities, which provides important context for understanding his clinical course and treatment approach.

Reviewer 2 Report

Comments and Suggestions for Authors

Case Report titled ,, Feasibility of Schema Therapy for Recurrent Depression in a Disaster Relief Worker with Prior Post-traumatic Stress Disorder Treatment Using Prolonged Exposure Therapy” seems scientifically interesting, giving a new highlight for therapy focused on and  addressed to  a disaster relief workers. In this paper schema therapy was  introduced to address over-adaptive work behaviors and vulnerabilities in personal relationships, which were identified as risk factors for relapse, even though ineffective prolonged exposure therapy was completed. As per authors ( quote) ,,  the combination of antidepressants and schema therapy led to sustained improvement where previous gains had deteriorated.” Thus, it seems the potential value of schema therapy as a secondary intervention when initial trauma-focused treatment gains are not maintained, especially in addressing the complex interplay between professional responsibilities and personal vulnerabilities in disaster relief workers facing prolonged challenges in disaster-affected communities, and treatment is continuing over 12 years. The reviewer suggests creating a "Strengths and Limitations" section in this text. However, in the opinion of the Reviewer, this is suspicious  using an AI language model (GPT-4)   for assistance with manuscript  editing and refinement.

Author Response

Case Report titled ,, Feasibility of Schema Therapy for Recurrent Depression in a Disaster Relief Worker with Prior Post-traumatic Stress Disorder Treatment Using Prolonged Exposure Therapy” seems scientifically interesting, giving a new highlight for therapy focused on and  addressed to  a disaster relief workers. In this paper schema therapy was  introduced to address over-adaptive work behaviors and vulnerabilities in personal relationships, which were identified as risk factors for relapse, even though ineffective prolonged exposure therapy was completed. As per authors ( quote) ,,  the combination of antidepressants and schema therapy led to sustained improvement where previous gains had deteriorated.” Thus, it seems the potential value of schema therapy as a secondary intervention when initial trauma-focused treatment gains are not maintained, especially in addressing the complex interplay between professional responsibilities and personal vulnerabilities in disaster relief workers facing prolonged challenges in disaster-affected communities, and treatment is continuing over 12 years. The reviewer suggests creating a "Strengths and Limitations" section in this text. However, in the opinion of the Reviewer, this is suspicious  using an AI language model (GPT-4)   for assistance with manuscript  editing and refinement.

Response:

Thank you for your thoughtful review and suggestions. We sincerely appreciate your recognition of our study's contribution to understanding treatment approaches for disaster relief workers.

Following your suggestion, we have added a dedicated "Limitations" section to the manuscript, while also maintaining discussion of the study's strengths throughout. The limitations section specifically addresses the methodological constraints of our single case study approach, the lack of systematic psychometric assessments, and practical implementation challenges of schema therapy. We believe this addition provides readers with a more balanced perspective on our findings while maintaining the paper's focus on the valuable clinical insights gained from this case.

Regarding your concerns about AI usage: We understand your attention to this important issue. We have been fully transparent about using GPT-4 for manuscript editing and refinement, as noted in our acknowledgments, which aligns with current publication practices. Many publishers now actively encourage such transparency in AI use for manuscript enhancement. We want to emphasize that all clinical observations, analyses, and conclusions were drawn independently by the authors based on direct clinical experience. The AI tool was used solely to improve English language expression and readability, not for content generation or analysis.

Reviewer 3 Report

Comments and Suggestions for Authors

The topic of this article intrigued me, and I reviewed it with interest. I would like to thank the authors for sharing their experiences with a trauma survivor case. To improve this article and contribute to trauma studies, I offer a few suggestions:

  1. Terminology: In the DSM-5, "post-traumatic ..." is referred to as a single term (Posttraumatic). Ensuring consistent terminology throughout the text may improve clarity.

  2. Background: More explanation on the theoretical connection between schema therapy and PTSD would be beneficial. Was the client also suffering from developmental trauma? What was the rationale behind choosing schema therapy over other psychological interventions such as TF-CBT, ACT, and others? It would be helpful to reference studies on developmental trauma and the role of trauma-related schemas in predicting PTSD.

  3. Methodology: The methodology section could benefit from expansion. First, specify the research design. Second, provide more details on the assessment tools and data collection methods, including descriptions of the questionnaires used. Third, even though this is a case study, data analysis methods should be explained in more detail, as they are directly tied to the research design.

  4. Results: A graph depicting the client’s progress and symptom changes, ideally after every five sessions, would enhance the results section. Highlighting the trends in PTSD symptom changes over time would add valuable insights.

  5. Discussion: The discussion could be expanded to address the findings in greater depth. Readers are interested in understanding how schema therapy impacted PTSD—its advantages, challenges, and the client’s perception of the therapy. Do the authors recommend schema therapy for PTSD treatment? Is it cost-effective?

  6. Research Ethics: Since this study involves clinical intervention, it would be appropriate to obtain an ethics code from the IRB and mention it in the methodology section.

Author Response

The topic of this article intrigued me, and I reviewed it with interest. I would like to thank the authors for sharing their experiences with a trauma survivor case. To improve this article and contribute to trauma studies, I offer a few suggestions:

Response: Thank you for your thoughtful review and interest in our trauma survivor case study. We appreciate your suggestions for improving the manuscript.

Terminology: In the DSM-5, "post-traumatic ..." is referred to as a single term (Posttraumatic). Ensuring consistent terminology throughout the text may improve clarity.

Response: Thank you for this attention to terminology consistency. We will revise the manuscript to use "posttraumatic stress disorder" as a single term throughout, following DSM-5 conventions.

Background: More explanation on the theoretical connection between schema therapy and PTSD would be beneficial. Was the client also suffering from developmental trauma? What was the rationale behind choosing schema therapy over other psychological interventions such as TF-CBT, ACT, and others? It would be helpful to reference studies on developmental trauma and the role of trauma-related schemas in predicting PTSD.

Response: Thank you for your valuable suggestions regarding the theoretical background of schema therapy in trauma treatment. We have addressed these points by revising our Introduction section to better articulate the theoretical rationale. Specifically, we have expanded the explanation of schema therapy (shown in green) to include:

  1. Its unique position among evidence-based trauma treatments
  2. The theoretical basis for applying schema therapy when vulnerability patterns persist after initial trauma-focused treatments
  3. Its particular relevance to disaster relief workers whose professional coping patterns may interact with personal vulnerability factors

Additionally, as detailed in Section 3.5, we have structured our clinical observations into three distinct phases (Sessions 1-21, 22-40, and 41-60), documenting how the therapeutic process addressed both trauma responses and underlying coping patterns. This demonstrates the practical application of schema therapy's theoretical framework in addressing the complex interplay between professional adaptation and personal vulnerability factors.

Methodology: The methodology section could benefit from expansion. First, specify the research design. Second, provide more details on the assessment tools and data collection methods, including descriptions of the questionnaires used. Third, even though this is a case study, data analysis methods should be explained in more detail, as they are directly tied to the research design.

Response: Thank you for your comment about methodology. We have added a detailed methodology description at the beginning of the Case Presentation section, explaining our naturalistic, longitudinal observation approach and data collection methods through clinical documentation.

This case originated from a small clinical practice setting focused on individualized care rather than systematic research. While psychometric assessments were not routinely implemented due to resource limitations, we collected data through clinical observations, session notes, patient self-reports, treatment response documentation, and follow-up assessments, supplemented by family observations.

The clinical course spans approximately 12 years, with particular focus on 2021-2024. While we acknowledge the limitations of this approach, we believe it offers valuable insights into real-world clinical practice and treatment adaptation in resource-limited settings.

Additionally, we have explicitly addressed this limitation in our newly created "Limitations" section, stating: "Second, due to the naturalistic observation approach, we did not conduct systematic psychometric assessments, which could have provided quantitative evidence of symptom changes and treatment effectiveness."

Results: A graph depicting the client’s progress and symptom changes, ideally after every five sessions, would enhance the results section. Highlighting the trends in PTSD symptom changes over time would add valuable insights.

Response: Thank you for your suggestion regarding quantitative progress tracking. While systematic psychometric measurements were not implemented in this clinical setting, we have substantially revised Section 3.5 to provide a detailed qualitative analysis of the therapeutic progression across 60 sessions. This revision documents the distinct phases of treatment, specifically highlighting:

  1. Initial phase (Sessions 1-21): Development of cognitive understanding and recognition of childhood trauma patterns
  2. Intermediate phase (Sessions 22-40): Implementation of experiential techniques and observable changes in trauma processing
  3. Final phase (Sessions 41-60): Integration and modification of maladaptive coping modes, with particular attention to emotional reconnection

This structured documentation of therapeutic progression provides insights into the patient's clinical improvement while maintaining the naturalistic observation approach appropriate for a case report from a clinical practice setting.

Discussion: The discussion could be expanded to address the findings in greater depth. Readers are interested in understanding how schema therapy impacted PTSD—its advantages, challenges, and the client’s perception of the therapy. Do the authors recommend schema therapy for PTSD treatment? Is it cost-effective?

Response: Thank you for these important suggestions regarding schema therapy's clinical implications. We have substantially revised our Discussion section to provide a more comprehensive analysis of schema therapy's impact, particularly in the Fourth paragraph (shown in green).

We have expanded our discussion to address schema therapy's role in addressing persistent vulnerabilities, its practical advantages and challenges (including cost-effectiveness considerations), and specific treatment recommendations. The revised discussion provides a clearer understanding of when schema therapy might be most beneficial - particularly as a secondary intervention for cases involving early developmental issues that complicate trauma responses, rather than as a first-line treatment for PTSD without such underlying patterns.

Research Ethics: Since this study involves clinical intervention, it would be appropriate to obtain an ethics code from the IRB and mention it in the methodology section.

Response: Thank you for this comment regarding research ethics. We would like to clarify that this is a case report documenting routine clinical care, not an interventional study. According to the "Ethical Guidelines for Life Science and Medical Research Involving Human Subjects" jointly established by MEXT, MHLW, and METI of Japan, case reports that document standard clinical practice are exempt from IRB review requirements (https://www.mext.go.jp/lifescience/bioethics/files/pdf/n2373_01.pdf).

Following standard clinical case report practices, we obtained written informed consent from the patient for publication of this case report, which ensures ethical handling of patient information while maintaining scientific rigor in case documentation.
